# Geometric Uncertainty for Detecting and Correcting Hallucinations in LLMs

## Abstract

Large language models demonstrate impressive results across diverse tasks but are still known to hallucinate, generating linguistically plausible but incorrect answers to questions. Uncertainty quantification has been proposed as a strategy for hallucination detection, requiring estimates for both global uncertainty (attributed to a batch of responses) and local uncertainty (attributed to individual responses). While recent black-box approaches have shown some success, they often rely on disjoint heuristics or graph-theoretic approximations that lack a unified geometric interpretation. We introduce a geometric framework to address this, based on archetypal analysis of batches of responses sampled with only black-box model access. At the global level, we propose Geometric Volume, which measures the convex hull volume of archetypes derived from response embeddings. At the local level, we propose Geometric Suspicion, which leverages the spatial relationship between responses and these archetypes to rank reliability, enabling hallucination reduction through preferential response selection. Unlike prior methods that rely on discrete pairwise comparisons, our approach provides continuous semantic boundary points which have utility for attributing reliability to individual responses. Experiments show that our framework performs comparably to or better than prior methods on short form question-answering datasets, and achieves superior results on medical datasets where hallucinations carry particularly critical risks. We also provide theoretical justification by proving a link between convex hull volume and entropy.

## 1 Introduction

Large language models (LLMs) have achieved remarkable performance across diverse natural language processing tasks (Guo et al., 2025; Anthropic, 2025; Gemini Team, Google DeepMind, 2025; OpenAI, 2025) and are increasingly applied in areas such as medical diagnosis, law, and financial advice (Yang et al., 2025; Chen et al., 2024; Kong et al., 2024). Hallucinations, however, where models generate plausible but false or fabricated content, pose significant risks for adoption in high-stakes applications (Farquhar et al., 2024). Recent work, for example, finds GPT-4 hallucinating in 28.6% of reference generation tasks (Chelli et al., 2024).

Uncertainty quantification (UQ) methods have been proposed to detect when models are producing unreliable outputs (Liu et al., 2025; Xiong et al., 2024). Effective UQ can serve as a critical layer of security and transparency, enabling systems to flag problematic responses and helping users exercise caution when reliability is compromised (Farquhar et al., 2024). This capability is particularly crucial for applications such as healthcare and legal services where incorrect information can cause significant harm (Huang et al., 2025; Asgari et al., 2025; Latif, 2025).

UQ methods can generally be divided into two groups: sampling-based and hidden-state-based. Sampling-based methods generate multiple responses for a prompt and estimate uncertainty over the resulting batch, often requiring only black-box access. Hidden-state-based methods use intermediate activations to estimate uncertainty at the single-response level, which requires white-box access. Previous work also distinguishes between external and internal uncertainty, the former arising from ambiguous queries and the latter from insufficient model knowledge (Li et al., 2025).

We additionally distinguish between uncertainty estimates at the batch level, which we term *global*, and those at the individual response level, which we term *local*. Prior work has referred to the latter

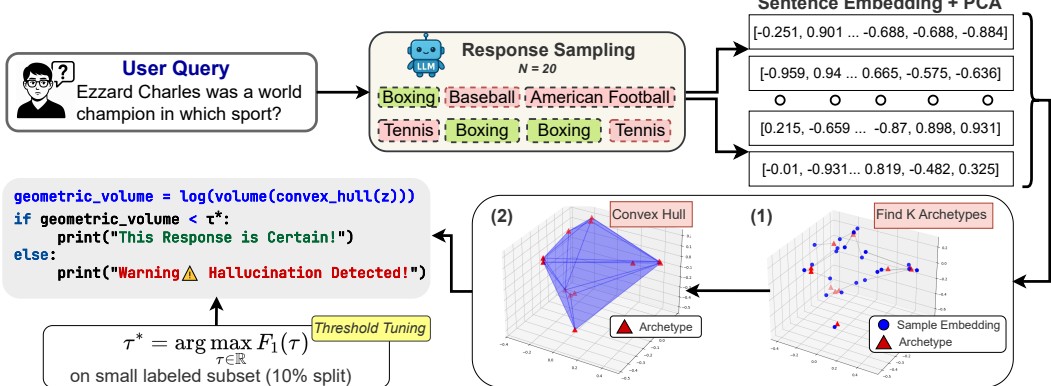

Figure 1: A schematic of geometric volume: (1) sample $n$ responses from the LLM, (2) embed and apply dimensionality reduction, (3) perform archetypal analysis and compute the convex hull, and (4) apply a threshold to detect hallucination.

as 'confidence' (Lin et al., 2024). A comprehensive UQ method should provide estimates of both types. In high-stakes settings, local UQ enables the selection of the most reliable response from several alternatives, thereby reducing the risk of harmful hallucinations. We argue that archetypal analysis uniquely enables this unification. The extremal archetypes define the global semantic boundaries and also serve as anchors for local response-level diagnostics, something no prior convex-hull or entropy-based method provides.

We introduce a geometric framework to quantify global and local uncertainty using only black-box access. Our approach is sampling-based: for a given prompt we generate a batch of responses at non-zero temperature and embed them with a sentence encoder. We then apply Archetypal Analysis (AA) (Cutler & Breiman, 1994) to identify a set of archetypes that span the embedding space. Our first contribution is *Geometric Volume*, a global uncertainty metric defined as the convex hull volume of the archetypes, which reflects the semantic spread of the batch.

Our second contribution is *Geometric Suspicion*, a local uncertainty measure that accounts for the distribution of responses within the spanned volume and how each response is reconstructed from the archetypes. By deriving local suspicion directly from the boundary conditions of the global volume, our method offers a geometrically grounded alternative to heuristic graph measures for differentiating high and low uncertainty responses.

We validate our framework on `CLAMBER`, `TriviaQA`, `ScienceQA`, `MedicalQA`, and `K-QA`. We also provide theoretical analysis linking convex hull volume to entropy. Our contributions are the following:

- We propose *Geometric Volume*, a global uncertainty metric that outperforms prior methods on medical QA and is competitive on standard benchmarks.

- We introduce *Geometric Suspicion*, a black-box, sampling-based local uncertainty method integrated within the same geometric framework, which reduces hallucinations by guiding Best-of-N response selection.

- We provide theoretical support by proving a link between the magnitude of the convex hull volume and the entropy of probability distributions defined within its boundaries.

## 2 RELATED WORKS

**Semantic Volume:** Our work is closely related to Semantic Volume (Li et al., 2025), which also analyzes the semantic dispersion of a batch of natural language outputs. Specifically, Semantic Volume computes the determinant of the Gram matrix formed from a batch of embeddings, where a low value ($\log \det(V^\top V)$) corresponds to low uncertainty and vice versa.

This value measures the volume of the parallelepiped formed by the embedding vectors. We suggest that our archetype-based method leads to a more accurate representation of the embedding space spanned by the embedded responses, whilst offering useful intermediate representations (i.e. the archetypes) for calculating local uncertainty.

**Convex Hull Approaches:** Several recent works use convex hull area over response embeddings as a proxy for uncertainty, projecting into two dimensions and summing hulls around clusters (Catak & Kuzlu, 2024; Catak et al., 2024). These works however vastly simplify the semantic space by projecting into only two dimensions with principal component analysis. They also first cluster responses before computing and summing the convex hull area around each cluster, which ignores how far apart in embedding space separate clusters may be. In doing so crucial information is lost regarding the variation in response meaning a model is likely to give to a certain prompt.

**Semantic Entropy and Self-Consistency:** Semantic entropy detects hallucinations by clustering semantically equivalent responses using bidirectional entailment, then computing entropy over those clusters (Farquhar et al., 2024). Self-consistency methods have built on their work and are emerging as a dominant paradigm (Taubenfeld et al., 2025; Wan et al., 2025; Savage et al., 2024), where multiple responses are sampled and their agreement is used as an uncertainty signal. Again, neither of these works offer additional methods for local UQ.

**Semantic Space Confidence:** Several approaches have been proposed to assess the local uncertainty, or confidence, of individual answers by analysis of the semantic answer embeddings. Semantic Density (Qiu & Miikkulainen, 2024) approximates the probability distribution over answers in semantic space and assigns high confidence to answers lying in high probability regions, but uses model likelihoods to do so and as such is not a black-box method. Lin et al. (2024) propose several black-box methods to assess global and local uncertainty, including *Degree* and *Eccentricity*. These methods use graphs computed from pairwise comparisons between answer representations, and as such lose the full semantic context of the response set, which is captured by our method.

**White-box Uncertainty:** Finally, numerous methods have been proposed to estimate uncertainty with white-box model access, using for instance token probabilities, logits or hidden layer activations without requiring additional sampling. These methods include minimum token probability, average token probability, and more advanced techniques that account for the semantic importance of individual tokens (Xia et al., 2025; Zhang et al., 2025; Liu et al., 2024; Malinin & Gales, 2020; Quevedo et al., 2024).

Unlike prior methods that collapse geometry into a single global score, our use of archetypal analysis yields interpretable anchor points that both define batch-level uncertainty and enable principled response-level attribution in a black-box setting. This enables fine-grained hallucination detection and improves interpretability without requiring access to internal model states.

## 3 METHODOLOGY

Our geometric framework quantifies LLM uncertainty through a two-tiered geometric analysis of response embeddings. First, it computes a global uncertainty score for a set of responses by modeling the geometric boundaries of their embeddings using archetypal analysis. Second, it introduces a local uncertainty score that ranks individual responses within a sampled batch, enabling a Best-of-N strategy that replaces hallucinated zero-temperature outputs with more reliable alternatives. This is achieved without access to internal model states, making it applicable to any black-box LLM.

### 3.1 PRELIMINARIES AND NOTATION

Let $q$ be an input query to an LLM. We first generate a default response, $r_{\text{default}}$, using greedy decoding. We then generate a set of $n$ responses $\{r_1, r_2, \ldots, r_n\}$ using a sampling strategy (e.g., non-zero temperature decoding). A pre-trained sentence embedding model (gte-Qwen2-1.5B-instruct) $\mathcal{E} : \mathcal{S} \to \mathbb{R}^d$ maps each response $r_i$ from the space of sentences $\mathcal{S}$ to a $d$-dimensional embedding vector $\mathbf{x}_i = \mathcal{E}(r_i)$. To address computational challenges associated with high-dimensional embeddings, we apply L2-normalization to each $\mathbf{x}_i$ and then use Principal Component Analysis (PCA) to

project the embeddings into a lower-dimensional subspace of dimension $d'$. The full set of responses is represented by the data matrix $\mathbf{X} \in \mathbb{R}^{n \times d'}$, where the $i$-th row is $\mathbf{x}_i^\top$.

## 3.2 GLOBAL UNCERTAINTY VIA GEOMETRIC VOLUME

At a high level, we use the set of responses $\mathbf{X}$ to evaluate the uncertainty associated with $r_{\text{default}}$. Our global uncertainty metric is based on archetypal analysis (AA) (Cutler & Breiman, 1994), an unsupervised method for finding representative *archetypes* that lie on the convex hull of the data. Unlike clustering methods that identify centroids within the data cloud, archetypes represent extremal points, thereby capturing the semantic boundaries of possible model responses.

**Archetypal Analysis Formulation**    Given a data matrix $\mathbf{X} \in \mathbb{R}^{n \times d'}$, the goal of Archetypal Analysis (AA) is to learn a set of $K$ archetypes that lie on the convex hull of the data. These archetypes form a dictionary $\mathbf{D} \in \mathbb{R}^{K \times d'}$, with each row representing an archetype. Each data point is required to be a convex combination of the archetypes, and at the same time, each archetype must itself be representable as a convex combination of the data points.

This bi-convex constraint leads to the following optimization problem (Abrol & Sharma, 2020):

$$\underset{\substack{\mathbf{B},\mathbf{A} \\ \mathbf{b}_j \in \Delta_n, \mathbf{a}_i \in \Delta_K}}{\arg\min} \quad \|\mathbf{X} - \mathbf{X}\mathbf{B}\mathbf{A}\|_F^2, \tag{1}$$

$$\Delta_n \triangleq [\mathbf{b}_j \succeq 0, \|\mathbf{b}_j\|_1 = 1], \Delta_K \triangleq [\mathbf{a}_i \succeq 0, \|\mathbf{a}_i\|_1 = 1].$$

Here, $\mathbf{a}_i$ and $\mathbf{b}_j$ are columns of $\mathbf{A} \in \mathbb{R}^{K \times n}$ and $\mathbf{B} \in \mathbb{R}^{n \times K}$, respectively. Thus, the archetype dictionary is given by $\mathbf{D} = \mathbf{B}^\top \mathbf{X}$.

In practice, Eq. 1 is solved iteratively via block-coordinate descent (Chen et al., 2014), alternating between updates of $\mathbf{A}$ and $\mathbf{B}$ until convergence. This procedure ensures that the resulting archetypes capture extremal directions of the data, approximating the vertices of the convex hull.

**Geometric Volume**    The learned archetypes $\mathbf{Z} = \{\mathbf{z}_1, \dots, \mathbf{z}_K\}$ lie on the boundary of the space spanned by the semantic embeddings of the response set. A high degree of uncertainty, or likely hallucination, manifests as a diverse set of responses, which geometrically corresponds to archetypes that are far apart in the embedding space. We capture this dispersion by computing the volume of the convex hull of the archetypes, $V = \text{volume}(\text{conv}(\mathbf{Z}))$. Our global uncertainty metric, termed Geometric Volume, is the logarithm of this volume:

$$H_G(\mathbf{X}) = \log\left(V + \epsilon\right) \tag{2}$$

where $\epsilon$ is a small constant (e.g., $10^{-12}$) for numerical stability. A larger volume indicates greater semantic dispersion among the sampled responses, which yields a higher Geometric Volume score and signals higher uncertainty.

## 3.3 SIMPLICIAL VOLUME AS A PROXY FOR UNCERTAINTY

To capture the geometric structure underlying LLM responses, we represent each response as a convex combination of learned archetypes, which lie at the vertices of a simplex in the embedding space. The response set lies within the simplex spanned by these archetypes, whose volume reflects the semantic dispersion involved in its generation. Intuitively, a larger simplex volume implies more "room" for diverse responses, geometrically constraining how concentrated the underlying distribution can be. This aligns with differential entropy, which quantifies the spread or unpredictability of a continuous distribution.

We now formalize this relationship with Theorem 1, which states that the volume of the archetypal simplex provides an upper bound on the differential entropy of any distribution supported within it, thereby linking geometric dispersion to semantic uncertainty. We provide the proof in the Appendix.

**Theorem 1.** *Let $\mathcal{A} = \{\mathbf{a}_1, \dots, \mathbf{a}_k\} \subset \mathbb{R}^d$ be a set of $k$ affinely independent archetypes. Let $\Delta = \text{conv}(\mathcal{A})$ denote the $(k-1)$-dimensional simplex they span, with intrinsic volume $V > 0$ measured using the $(k-1)$-dimensional Hausdorff measure $\mathcal{H}^{k-1}$ on the affine span of $\Delta$. Let $\mathbf{x}$ be*

*a continuous random vector supported on $\Delta$, with density $p(\mathbf{x})$ that is absolutely continuous with respect to $\mathcal{H}^{k-1}$. Then the differential entropy of $\mathbf{x}$ satisfies:*

$$H(\mathbf{x}) = - \int_{\Delta} p(\mathbf{x}) \log p(\mathbf{x}) \, d\mathcal{H}^{k-1}(\mathbf{x}) \leq \log V, \tag{3}$$

*and the upper bound is achieved if and only if $\mathbf{x}$ is uniformly distributed over $\Delta$.*

*Proof.* See Appendix A.1. $\square$

### 3.4 LOCAL UNCERTAINTY VIA GEOMETRIC SUSPICION

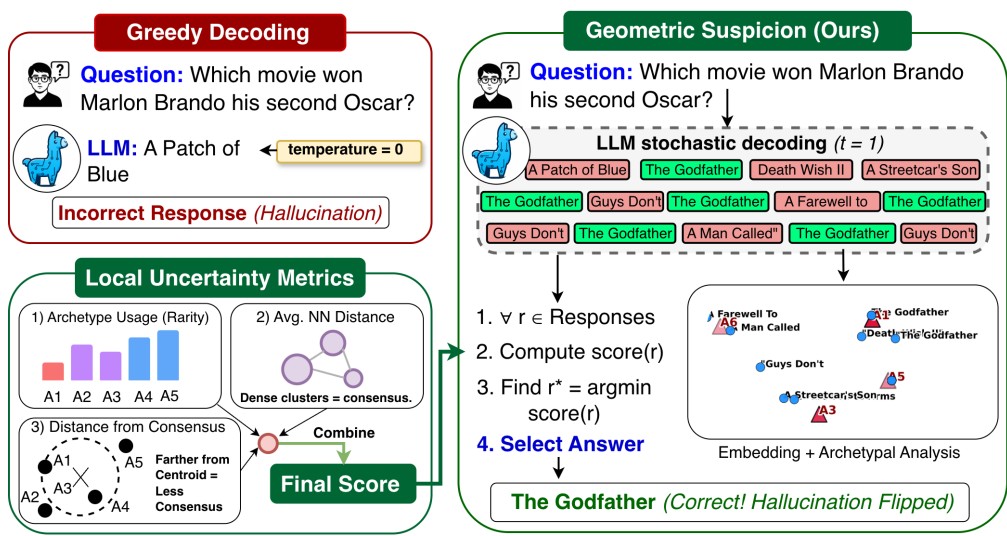

Figure 2: We demonstrate local uncertainty metrics using Archetype Rarity, Average Distance to Nearest Neighbors, and Distance from Consensus, which can transform hallucinated responses at temperature 0 into correct responses.

The global score $H_G(\mathbf{X})$ allows us to classify hallucinations successfully but only facilitates deferral as a method of reducing hallucinations, as it does not attribute uncertainty to individual responses within a set. Sensible local scores would enable hallucination reduction by preferentially selecting more certain responses from the response set. While recent graph-based approaches have demonstrated the utility of local metrics (Lin et al., 2024), we propose *Geometric Suspicion*, a local measure explicitly derived from the convex hull geometry used for our global metric. This is built from three per–response terms derived from the archetypal coefficients $\mathbf{A}$ and the answer embeddings $\mathbf{X}$.

**Local Density** For each response embedding $\mathbf{x}_i \in \mathbf{X}$, let $\mathcal{N}_k(\mathbf{x}_i)$ be its $k$-nearest neighbours. We score

$$L(r_i) = \frac{1}{k} \sum_{\mathbf{x}_j \in \mathcal{N}_k(\mathbf{x}_i)} \left\| \mathbf{x}_i - \mathbf{x}_j \right\|_2, \tag{4}$$

so that isolated (locally sparse) responses receive larger values.

**Distance from Consensus** To place each response in a global geometric context, we measure the distance to the batch consensus

$$D(r_i) = \left\| \mathbf{x}_i - \mathbf{x}_c \right\|_2, \qquad \mathbf{x}_c = \frac{1}{n} \sum_{j=1}^{n} \mathbf{x}_j. \tag{5}$$

Large values indicate semantic deviation from the central tendency of the answer set.

**Usage Rarity** The $i$-th row of $\mathbf{A}$, denoted $\boldsymbol{\alpha}_i = [A_{i1}, \ldots, A_{iK}]$ contains coefficients describing how the response embedding $\mathbf{x}_i$ is reconstructed from the set of learned archetypes $\mathbf{Z}$. The score for a response $r_i$ is high if its reconstruction coefficients $A_{ik}$ depend heavily on archetypes that are, on average, rarely used to explain the rest of the data.:

$$U(r_i) = \sum_{k=1}^{K} A_{ik}\big(1 - \bar{\alpha}_k\big), \quad \text{where} \quad \bar{\alpha}_k = \frac{1}{n}\sum_{j=1}^{n} A_{jk}. \tag{6}$$

**Final Suspicion Score (Fisher combination)** We convert each per–metric score into a within–batch empirical right–tail $p$–value. For metric $M \in \{L, D, U\}$ and response $r_i$,

$$p_i^{(M)} = \frac{1 + \Big|\big\{\, j \in \{1, \ldots, n\}: \; s_j^{(M)} \geq s_i^{(M)} \,\big\}\Big|}{n+1}, \tag{7}$$

where $s_i^{(M)}$ is oriented so that larger values indicate greater suspicion. We then combine the three $p$–values using Fisher's method to obtain the final suspicion statistic

$$S(r_i) = -2 \sum_{M \in \{L, D, U\}} \log p_i^{(M)}. \tag{8}$$

**Relation to prior local UQ baselines and intuition** Our two geometric terms align closely with established graph-based baselines, namely Degree and Eccentricity (Lin et al., 2024). *Local Density* is a pointwise analogue of *Degree* on an answer–answer affinity graph: degree is a (monotone) transform of local similarity mass, while equation 4 measures the corresponding local spacing in $\mathbf{X}$. *Distance from Consensus* mirrors *Eccentricity* (distance to a central embedding in a low-dimensional spectral space): both penalize globally off-centre responses, even when they form dense clusters. *Usage Rarity* contributes complementary geometric signal from archetypal analysis by highlighting reliance on rarely used (hull-boundary) archetypes, which our global theory links to increased dispersion. Together, the three terms provide local sparsity (degree–like), global deviation (eccentricity–like), and boundary reliance (archetypal) evidence.

To derive and support our specific choice of terms, we conduct an analysis of the datasets generated for our global uncertainty experiments, which can be found in Appendix Section D. In Figure 3, we show examples where component metrics used in isolation would not select an optimal response from a batch, whereas their combination overcomes individual limitations. Degree / Local Density used in isolation, for instance, struggle in situations where there exist multiple dense clusters in the semantic response set.

### 3.5 HALLUCINATION DETECTION

Our geometric volume metric can be used to detect hallucinations at the response set level. Given $r_{\text{default}}$ and a corresponding set of responses with embedding matrix $\mathbf{X}$, we classify $r_{\text{default}}$ as hallucinated if the global uncertainty exceeds a threshold $\tau$, i.e., if $H_G(\mathbf{X}) > \tau$. The threshold $\tau$ is a tunable hyperparameter, which can be optimized on a small labeled validation set to maximize a classification objective such as F1-score.

### 3.6 HALLUCINATION CORRECTION

Beyond merely identifying uncertain responses, the primary application of our suspicion score $S(r)$ is to actively improve model reliability. We employ it in a Best-of-N (BoN) sampling framework, where the goal is to select the most plausible response from a set of $n$ candidates generated for a given question.

Given a set of $n$ sampled responses $R = \{r_1, ..., r_n\}$ for a single prompt, our uncertainty-guided selection process chooses the single response, $r^*$, that exhibits the minimum suspicion score. This approach replaces the model's default (and potentially hallucinated) answer with one deemed most reliable by our proposed metric.

To quantify the efficacy of this selection strategy, we measure the net reduction in the hallucination rate across a dataset. First, we define the baseline hallucination rate, $H_{\text{baseline}}$, as the proportion of questions in a test set for which the model's default answer $r_{\text{default}}$ was a hallucination.

Next, we define the post-selection hallucination rate, $H_{\text{BoN-U}}$, as the proportion of questions where the answer $r^*$ selected by our uncertainty-guided method is a hallucination.

The overall improvement is captured by the Delta Hallucination Rate, $\Delta H$, which is the absolute reduction in the hallucination rate achieved by our method.

$$\Delta H = H_{\text{baseline}} - H_{\text{BoN-U}}. \qquad (9)$$

A positive $\Delta H$ indicates that the suspicion score is successfully guiding the selection towards non-hallucinated responses, thereby demonstrating the practical utility of the local uncertainty measure.

## 4 EXPERIMENTS

We conduct a series of experiments to evaluate the effectiveness of our geometric framework for hallucination detection. We assess its performance under external uncertainty (prompt ambiguity) and internal uncertainty (fact-checking). We also evaluate uncertainty methods in more realistic, long-form medical question-answering scenarios that better reflect real-world conditions.

### 4.1 BENCHMARKS AND BASELINES

We evaluate on five benchmarks covering both external and internal uncertainty: CLAMBER (ambiguous prompts), TriviaQA (short-form QA), ScienceQA (scientific reasoning), MedicalQA (high-stakes medical QA), and K-QA (real-world medical scenarios). These benchmarks span ambiguous queries, factual QA, and long-form medical domains. Dataset construction details are provided in Section B of the Appendix. We compare our method with Semantic Volume (Li et al., 2025), Semantic Entropy (Farquhar et al., 2024), and P(true), which estimates the probability that a model's generated response is correct by directly asking the model itself to self-assess its confidence (Lin et al., 2024).

We evaluate our local uncertainty quantification methods on the same datasets curated for the global Internal Uncertainty experiments. We first filter each dataset to only contain questions where the set of sampled responses contains examples of both hallucinations and non-hallucinations. For each sample in the perturbed responses, we then compute the local uncertainty score using the archetypal reconstruction matrix $\mathbf{A}$ and embeddings $\mathbf{X}$. We compare our method with Degree and Eccentricity (Lin et al., 2024), which are based on graphs derived from pairwise response comparisons. We report $\Delta H$ and AUARC (Area Under the Accuracy–Rejection Curve). The latter is computed per question by ranking responses by uncertainty, progressively rejecting the top-$k$, measuring accuracy on the retained set, and averaging over all rejection rates ($k = 0, \ldots, n - 1$). A random predictor yields AUARC equal to the base accuracy.

## 5 RESULTS

We evaluate our geometric framework for hallucination detection across multiple benchmarks and models. For global uncertainty, we test Geometric Volume's ability to classify $r_{\text{default}}$ as reliable or hallucinated given a response set. For local uncertainty, we assess the Geometric Suspicion score in reducing hallucinations via a Best-of-N selection strategy. Each experiment is repeated three times to mitigate sampling stochasticity, and we report the mean and standard deviation of all metrics.

### 5.1 GLOBAL UNCERTAINTY DETECTION

Geometric Volume shows strong performance across all benchmarks and models investigated. On K-QA, it significantly outperforms baselines, achieving the best F1 Score and AUROC with GPT-3.5-Turbo (75.4 and 67.8) and the strongest AUROC with GPT-4o Mini (67.7) and Llama3.1-8b (72.8). On MedicalQA, it remains highly competitive, delivering top F1 Scores with GPT-3.5-Turbo (73.9) and Qwen3-8b (75.2), as well as the best AUROC with GPT-4o Mini (60.9) (Table 1).

On CLAMBER, which tests detection of ambiguous prompts, Geometric Volume achieves the highest AUROC with GPT-4o Mini (61.7) and Qwen3-8b (59.1), while GPT-3.5-Turbo and Llama3.1-8b reach the highest F1 Scores. In TriviaQA, it results in the highest F1 and AUROC with GPT-4o

Table 1: Results across five benchmarks and four LLMs comparing external uncertainty estimation methods. Reported values are mean$_{std}$. Blue cells indicate best results from **Ours**, red cells indicate best results from baselines. AUC = AUROC.

| Method | CLAMBER F1 | CLAMBER AUC | TriviaQA F1 | TriviaQA AUC | ScienceQA F1 | ScienceQA AUC | MedicalQA F1 | MedicalQA AUC | K-QA F1 | K-QA AUC |
|---|---|---|---|---|---|---|---|---|---|---|
| **GPT-4o Mini** | | | | | | | | | | |
| $p(\text{true})$ | $47.0_{1.6}$ | $58.5_{0.4}$ | $59.2_{3.6}$ | $65.8_{4.0}$ | $53.0_{4.0}$ | $77.6_{3.4}$ | $73.6_{1.2}$ | $59.6_{1.0}$ | $61.5_{4.7}$ | $63.4_{1.5}$ |
| Sem. Entropy | $63.9_{0.3}$ | $48.1_{0.8}$ | $67.7_{3.4}$ | $75.3_{0.3}$ | $48.6_{2.7}$ | $65.3_{1.6}$ | $70.9_{1.0}$ | $59.3_{1.0}$ | $60.4_{4.8}$ | $50.5_{5.7}$ |
| Sem. Volume | $68.5_{0.4}$ | $55.8_{0.9}$ | $69.9_{1.0}$ | $74.4_{0.3}$ | $54.0_{4.4}$ | $57.9_{1.5}$ | $73.0_{1.2}$ | $59.4_{1.0}$ | $68.2_{1.6}$ | $65.2_{1.1}$ |
| **Ours** | $66.1_{0.3}$ | $61.7_{0.7}$ | $71.1_{1.4}$ | $76.7_{0.3}$ | $58.7_{3.0}$ | $68.8_{2.2}$ | $73.6_{1.2}$ | $60.9_{1.4}$ | $68.6_{0.8}$ | $67.7_{1.3}$ |
| **GPT-3.5-Turbo** | | | | | | | | | | |
| $p(\text{true})$ | $53.0_{0.5}$ | $60.1_{0.6}$ | $49.6_{1.0}$ | $68.2_{2.7}$ | $57.0_{5.6}$ | $76.1_{1.5}$ | $60.0_{11.1}$ | $71.9_{1.9}$ | $71.8_{1.4}$ | $60.5_{1.2}$ |
| Sem. Entropy | $66.6_{0.2}$ | $53.8_{0.2}$ | $69.6_{1.2}$ | $75.6_{0.4}$ | $62.3_{3.3}$ | $69.2_{1.5}$ | $71.3_{1.5}$ | $68.7_{0.3}$ | $40.3_{6.9}$ | $66.4_{1.2}$ |
| Sem. Volume | $68.5_{0.1}$ | $54.4_{0.6}$ | $71.7_{0.6}$ | $74.9_{0.8}$ | $64.3_{3.2}$ | $62.7_{4.5}$ | $73.1_{0.3}$ | $61.7_{0.6}$ | $74.8_{1.0}$ | $65.3_{0.7}$ |
| **Ours** | $82.5_{2.6}$ | $61.2_{1.3}$ | $71.8_{1.5}$ | $76.7_{0.4}$ | $70.1_{3.7}$ | $73.4_{3.6}$ | $73.9_{1.1}$ | $64.9_{0.4}$ | $75.4_{0.4}$ | $67.8_{3.2}$ |
| **Qwen3-8b** | | | | | | | | | | |
| $p(\text{true})$ | $55.0_{0.8}$ | $57.1_{0.3}$ | $64.1_{3.6}$ | $86.0_{1.1}$ | $41.3_{4.3}$ | $66.6_{1.2}$ | $74.4_{1.3}$ | $66.9_{1.7}$ | $72.8_{0.8}$ | $66.3_{1.4}$ |
| Sem. Entropy | $63.8_{0.2}$ | $49.3_{0.5}$ | $78.1_{0.6}$ | $84.2_{0.3}$ | $22.2_{19.3}$ | $58.9_{0.8}$ | $73.2_{0.5}$ | $71.1_{0.6}$ | $73.4_{1.1}$ | $72.5_{0.8}$ |
| Sem. Volume | $67.1_{0.2}$ | $55.3_{0.2}$ | $74.4_{1.4}$ | $80.8_{0.7}$ | $66.6_{0.1}$ | $48.1_{0.6}$ | $74.4_{0.8}$ | $69.1_{0.3}$ | $73.3_{1.1}$ | $69.8_{0.2}$ |
| **Ours** | $65.9_{0.1}$ | $59.1_{0.4}$ | $76.9_{0.3}$ | $82.1_{0.4}$ | $66.7_{0.2}$ | $67.2_{0.4}$ | $75.2_{1.1}$ | $69.5_{0.6}$ | $73.6_{1.2}$ | $69.8_{0.2}$ |
| **Llama3.1-8b** | | | | | | | | | | |
| $p(\text{true})$ | $62.8_{2.0}$ | $62.0_{0.5}$ | $60.8_{2.7}$ | $58.2_{12.2}$ | $45.3_{2.9}$ | $69.1_{2.7}$ | $74.4_{2.2}$ | $76.4_{2.1}$ | $55.8_{3.8}$ | $67.9_{4.0}$ |
| Sem. Entropy | $67.1_{0.4}$ | $58.0_{0.7}$ | $71.0_{1.0}$ | $76.9_{0.6}$ | $73.0_{0.8}$ | $77.0_{0.8}$ | $76.6_{0.7}$ | $72.0_{1.2}$ | $82.8_{1.9}$ | $69.9_{2.5}$ |
| Sem. Volume | $66.8_{0.1}$ | $51.0_{0.4}$ | $70.6_{0.0}$ | $72.7_{0.7}$ | $66.4_{0.2}$ | $75.4_{0.7}$ | $75.4_{0.2}$ | $56.2_{0.7}$ | $81.7_{0.4}$ | $65.0_{2.2}$ |
| **Ours** | $76.2_{2.1}$ | $54.9_{1.2}$ | $71.3_{0.5}$ | $75.4_{0.5}$ | $78.6_{0.7}$ | $81.6_{0.4}$ | $75.3_{0.2}$ | $67.2_{0.9}$ | $82.7_{0.1}$ | $72.8_{1.1}$ |

Mini (71.1, 76.7) and GPT-3.5-Turbo (71.8 F1, 76.7 AUROC). On ScienceQA, it shows particular strength with Llama3.1-8b, where it delivers 78.6 F1 and 81.6 AUROC. These results highlight its ability to model semantic dispersion effectively.

## 5.2 REDUCING HALLUCINATION RATES WITH LOCAL UNCERTAINTY

We show in Table 2 that using our local uncertainty metric, we are able to reduce hallucination rates across all models and datasets investigated. In general it surpasses baselines across models and datasets, with particularly dominant performance in the MedicalQA task.

We attribute this performance primarily due to the combination of terms employed, compared to baselines which use only one term. In Figure 3 we show how reliance on one notion of 'confidence', such as local density or distance from consensus, would lead to Best-of-N selection of a hallucinated response, whereas our composite metric selects a correct response.

Figure 3(a) shows a straightforward example where all three of the component terms have high p-values. The correct answer is in a dense local cluster, near a global consensus point, and can be reconstructed using commonly used archetypes. Figure 3(b) shows a slightly harder example, where there are multiple clusters of semantically similar answers. Here the desired answer is not within the densest local cluster or the closest to a global consensus, but has lower suspicion due to its non-reliance on rarely used archetypes. The example in Figure 3(c) can be used to understand the necessity of both local density and distance from consensus metrics. Here we have a few dense clusters, resulting in high local density for the majority of the batch. The global semantic consensus point however is closer to the 'Indonesia' cluster than it is the 'Not specified' or 'Unknown entity' clusters. This is because the hallucinated answers of 'Brazil' or 'Pakistan' are countries and semantically closer to the correct answer, moving the global consensus further in this direction. Finally, Figure 3(d) shows our framework can be useful in cases of diverse answer batches without dense

Table 2: Comparison of local uncertainty methods on $\Delta H$ (left) and AUARC (right). Values are percentages; subscripts denote standard deviation across runs. Blue cells indicate the best value among **Ours**; red cells indicate the best baseline.

| Method | TriviaQA | | ScienceQA | | MedicalQA | | K-QA | |
|---|---|---|---|---|---|---|---|---|
| Method | $\Delta H$ | AUARC | $\Delta H$ | AUARC | $\Delta H$ | AUARC | $\Delta H$ | AUARC |
| **GPT-4o Mini** | | | | | | | | |
| Eccentricity | $-1.0_{1.1}$ | $48.3_{1.0}$ | $4.9_{0.9}$ | $16.4_{4.9}$ | $2.7_{1.0}$ | $55.7_{2.1}$ | $4.9_{4.0}$ | $58.5_{0.2}$ |
| Degree | $7.7_{2.5}$ | $51.9_{0.3}$ | $5.7_{0.6}$ | $16.8_{4.6}$ | $7.0_{3.7}$ | $57.8_{2.4}$ | $15.1_{1.9}$ | $63.7_{0.3}$ |
| **Ours** | $6.9_{1.1}$ | $51.9_{0.2}$ | $6.2_{1.0}$ | $17.3_{4.8}$ | $8.9_{1.6}$ | $58.6_{1.8}$ | $15.3_{2.9}$ | $64.1_{0.2}$ |
| **GPT-3.5-Turbo** | | | | | | | | |
| Eccentricity | $1.0_{2.0}$ | $55.8_{0.2}$ | $5.5_{1.4}$ | $25.8_{5.7}$ | $1.7_{2.0}$ | $51.5_{1.2}$ | $20.8_{4.5}$ | $64.3_{0.2}$ |
| Degree | $5.7_{0.8}$ | $58.1_{1.0}$ | $4.5_{2.9}$ | $25.1_{6.6}$ | $8.7_{1.4}$ | $54.4_{0.6}$ | $32.2_{3.2}$ | $68.8_{1.6}$ |
| **Ours** | $6.0_{0.6}$ | $58.1_{0.7}$ | $2.8_{2.5}$ | $24.5_{6.8}$ | $8.8_{0.6}$ | $55.0_{0.5}$ | $31.3_{3.0}$ | $68.3_{1.7}$ |
| **Qwen3-8b** | | | | | | | | |
| Eccentricity | $-9.2_{1.8}$ | $47.3_{0.9}$ | $2.2_{2.3}$ | $13.2_{0.9}$ | $-1.0_{1.6}$ | $49.7_{2.9}$ | $9.8_{4.8}$ | $55.2_{2.8}$ |
| Degree | $-0.7_{1.6}$ | $51.6_{0.8}$ | $3.7_{3.3}$ | $13.7_{0.1}$ | $7.5_{2.2}$ | $53.4_{2.3}$ | $15.3_{8.6}$ | $58.5_{3.6}$ |
| **Ours** | $-0.5_{0.7}$ | $51.6_{0.8}$ | $3.7_{3.3}$ | $14.8_{0.5}$ | $8.0_{1.1}$ | $54.1_{2.4}$ | $19.0_{7.2}$ | $59.9_{3.2}$ |
| **Llama3.1-8b** | | | | | | | | |
| Eccentricity | $-2.6_{0.3}$ | $42.2_{0.2}$ | $8.2_{1.8}$ | $25.0_{1.6}$ | $-7.5_{2.9}$ | $41.8_{0.8}$ | $9.2_{3.9}$ | $44.8_{0.6}$ |
| Degree | $4.5_{2.1}$ | $45.5_{0.8}$ | $7.0_{3.0}$ | $24.5_{2.1}$ | $4.0_{3.0}$ | $47.2_{0.8}$ | $12.3_{3.3}$ | $46.4_{1.6}$ |
| **Ours** | $2.1_{1.7}$ | $44.7_{0.6}$ | $8.0_{2.9}$ | $25.5_{2.0}$ | $5.2_{1.2}$ | $48.6_{1.3}$ | $15.3_{3.6}$ | $46.9_{1.3}$ |

local clusters. In this example approximately half of the answers are semantically unique, but the use of commonly used archetypes in the 'Michael Nesmith' cluster allows us to find a low suspicion example which proves to be correct.

An interesting case is Qwen3-8B for the TriviaQA dataset, where all methods tested fail to reduce the hallucination rate. We find that compared to other model/dataset combinations, in this case the model is more often *confidently wrong* when it hallucinates: that is, very few of the sampled answers are non-hallucinations, so choosing an optimal answer becomes a difficult task. We comment on this method limitation in Section 6 and provide the evidence for this conclusion in Appendix Section E.

# 6 DISCUSSION

Our geometric framework views language model UQ as modelling a probability distribution over answers in semantic space. The entropy of this distribution, bounded by geometric volume, yields useful global uncertainty signals. Like previous methods such as Semantic Density, we then frame local UQ as finding optimal points within the distribution. Our convex hull approach is a novel method for defining the semantic distribution boundaries, whilst our local metric approximates a 'confidence' distribution within the boundaries without access to model likelihoods.

Our framework has several limitations, which we aim to address in future work. Firstly, it relies on the capability of a single embedding vector to sufficiently capture the semantic content of an answer. As answers get longer and semantically more complex, this assumption may break down. Recent work has investigated this and suggests that the principle of semantic distribution analysis with a single embedding vector holds promise for answers of up to 1000 words (Bhardwaj et al., 2025), but it remains to be proven for complex datasets beyond QA tasks.

Our method relies on the principle of semantic dispersion in sampled answers predicting hallucination. It is therefore most effective in settings where the model is uncertain and wrong, i.e. there is sufficient diversity in the sampled answers. There exist cases where the model is confident and wrong (no diversity in sampled answers). White-box methods such as factuality probes (Han et al., 2025) may be able to address these cases, but to our knowledge no black-box methods have successfully solved this problem.

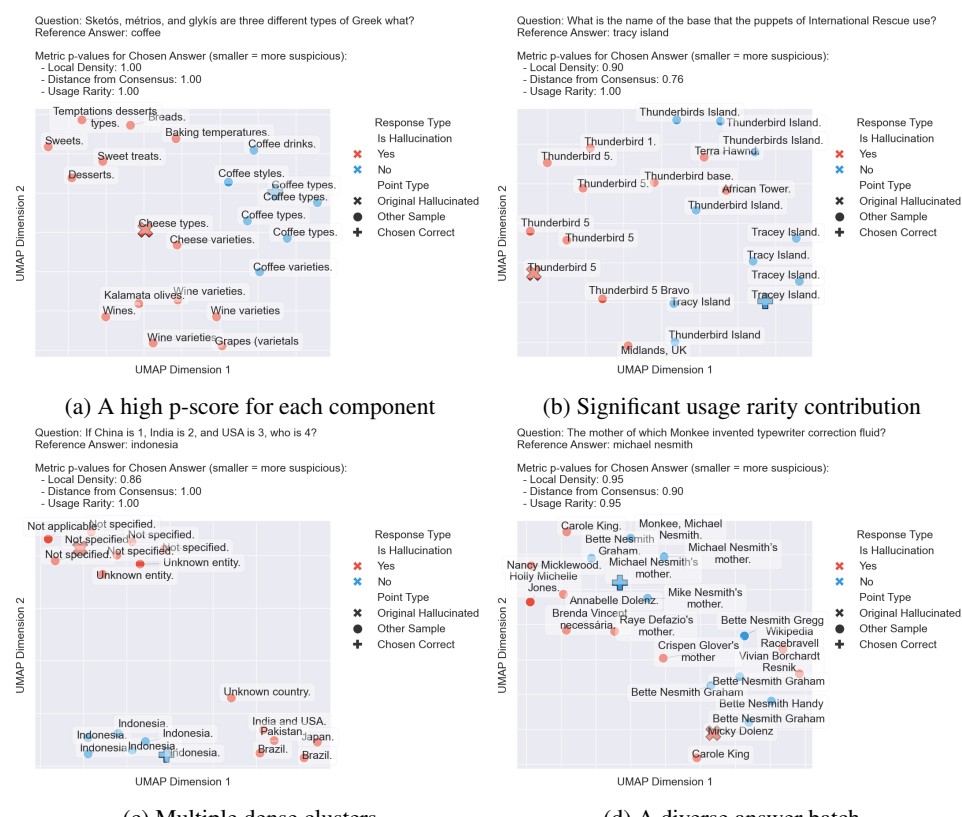

Figure 3: UMAP plots for cases where we were able to correct hallucinations with our local uncertainty metric, using the TriviaQA dataset and GPT-4o-mini model. In each plot, the red X shows $r_{\text{default}}$, which was determined to be a hallucination by a judge LLM. The other points show the batch of $n = 20$ answers sampled from the same model with a temperature of one. The blue cross shows the answer selected by our framework (i.e. with lowest suspicion), which was determined by the same judge LLM to be a non-hallucination.

## 7    CONCLUSION

We presented a geometric framework for detecting and mitigating hallucinations in large language models using only black-box access. At the global level, Geometric Volume quantifies semantic dispersion through convex hull analysis of archetypes and provides a principled proxy for uncertainty. At the local level, Geometric Suspicion identifies unreliable responses within a batch and enables a Best-of-N selection strategy that consistently reduces hallucination rates across benchmarks. Our experiments demonstrate strong performance in both standard QA tasks and high-stakes medical domains, while our theoretical analysis establishes a direct connection between convex geometry and entropy. These results highlight geometry as a powerful tool for linking global dispersion with local reliability, and open new directions for uncertainty estimation that are interpretable, data-efficient, and broadly applicable.

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

# A  THEORETICAL ANALYSIS

We now present a theoretical analysis to formally justify the different components for the proposed uncertainty estimation framework.

**Corollary 1.** *Let a response be represented as a convex combination of archetypes lying in a simplex $\Delta \subset \mathbb{R}^d$ with volume $V > 0$. Then the differential entropy of $\mathbf{x}$, with density $p(\mathbf{x})$ supported on $\Delta$, satisfies:*

$$H(\mathbf{x}) \leq \log V. \tag{10}$$

*Consequently, the volume $V$ provides a geometric upper bound on the differential entropy of a response distribution defined within the simplex. Differential entropy in this context is analogous to the semantic uncertainty of a response set. This bound is computationally efficient to estimate and can be used to detect low-confidence or out-of-distribution inputs without explicitly modeling $p(\mathbf{x})$.*

This result follows directly from Theorem 1, by applying the entropy–volume bound to the simplex associated with the response's archetypal representation.

## A.1  PROOF OF THEOREM 1

*Proof.* Let $u(\mathbf{x}) = \frac{1}{V}$ denote the uniform distribution over $\Delta$, where $V$ is the volume of $\Delta$ under the $(k-1)$-dimensional Hausdorff measure $\mathcal{H}^{k-1}$. The differential entropy of $u$ is:

$$H(u) = -\int_\Delta \frac{1}{V} \log\left(\frac{1}{V}\right) d\mathcal{H}^{k-1}(\mathbf{x}) = \log V, \tag{11}$$

since $\int_\Delta d\mathcal{H}^{k-1}(\mathbf{x}) = V$.

The Kullback–Leibler (KL) divergence from $p$ to $u$ is:

$$D_{\mathrm{KL}}(p\|u) = \int_\Delta p(\mathbf{x}) \log\left(\frac{p(\mathbf{x})}{u(\mathbf{x})}\right) d\mathcal{H}^{k-1}(\mathbf{x}). \tag{12}$$

Substituting $u(\mathbf{x}) = \frac{1}{V}$ yields:

$$D_{\mathrm{KL}}(p\|u) = \int_\Delta p(\mathbf{x}) \log p(\mathbf{x}) \, d\mathcal{H}^{k-1}(\mathbf{x}) + \log V. \tag{13}$$

Rearranging terms, we obtain:

$$H(\mathbf{x}) = -\int_\Delta p(\mathbf{x}) \log p(\mathbf{x}) \, d\mathcal{H}^{k-1}(\mathbf{x}) = \log V - D_{\mathrm{KL}}(p\|u). \tag{14}$$

By the non-negativity of KL divergence (Gibbs' inequality), $D_{\mathrm{KL}}(p\|u) \geq 0$, with equality if and only if $p = u$ everywhere. Thus:

$$H(\mathbf{x}) \leq \log V, \tag{15}$$

with equality if and only if $\mathbf{x}$ is uniformly distributed over $\Delta$. $\qquad\square$

# B  EXPERIMENTAL SETUP

## B.1  BENCHMARKS AND BASELINES

**CLAMBER (External Uncertainty).**   To evaluate the detection of external hallucinations arising from ambiguous user prompts, we use the CLAMBER benchmark (Zhang et al., 2024). We use 3,202 queries from the dataset, where the model's uncertainty should reflect the ambiguity of the input rather than a lack of internal knowledge.

**TriviaQA (Internal Uncertainty).**   We also evaluate on TriviaQA, a short-form question answering benchmark (Joshi et al., 2017). From the TriviaQA dataset, we construct a balanced test set of 1,000 samples, consisting of 500 hallucinated and 500 non-hallucinated examples.

**ScienceQA Multiple Choice QA**  We also used ScienceQA as a benchmark (Lu et al., 2022), which is a multimodal multiple-choice dataset designed to assess scientific reasoning across natural, social, and language sciences. For our evaluation, we filtered the test set to include only text-only questions and created a balanced benchmark of 200 hallucinating and 200 non-hallucinating examples for each LLM.

**MedicalQA (High-Stakes Domain).**  To assess performance in a critical, high-stakes domain, we use a medical question-answering benchmark subset (500 examples) from MedQA and MedMcQA (Zhang et al., 2023). This dataset tests the model's ability to provide factually correct answers to medical questions, where hallucinations carry significant risk. MedicalQA is a long-form question answering benchmark, unlike TriviaQA, and provides deeper insight into hallucination in real-world applications.

**K-QA (Real-World Medical Scenarios).**  Finally, to evaluate uncertainty in real-world medical scenarios, we leverage 201 real patient questions from the K-QA dataset (Manes et al., 2024), a long-form generation benchmark with expert-written answers reviewed by licensed physicians. These detailed responses serve as high-quality references for assessing the accuracy and reliability of model outputs.

## B.2 DATASET CURATION

**External Uncertainty.**  For the CLAMBER benchmark, we evaluate each of the 3,202 queries, where each $q_i$ is annotated with a binary label $y_i \in 0, 1$ indicating whether it is ambiguous. We generate $n = 20$ perturbations $\{q_i^{(j)}\}_{j=1}^n$ for each query by prompting each LLM following previous work (Li et al., 2025), and obtain embeddings $\{x_i^{(j)}\}_{j=1}^n$ via the Alibaba-NLP/gte-Qwen2-1.5B-instruct model.

**Internal Uncertainty.**  To create hallucination detection datasets from traditional QA benchmarks, we convert question–answer pairs $(q_i, a_i^{\text{ref}})$ into labeled examples. For each question $q_i$, we generate a response $r_i = \text{LLM}(q_i)$ using LLM. We then assign a hallucination label $y_i \in \{0, 1\}$ by comparing $r_i$ to the reference answer $a_i^{\text{ref}}$. In high-stakes domains such as medical QA, this comparison is performed by an LLM-as-a-judge (Gu et al., 2024) (GPT-4o) using expert-annotated references. For example, given a real-world clinical question and its verified answer, we label $r_i$ as hallucinated if the LLM judge determines it deviates from the reference. For datasets like TriviaQA, where answers are typically short (1–3 words), we instead compute ROUGE-L F1 and label $r_i$ as hallucinated if $\text{ROUGE}(r_i, a_i^{\text{ref}}) < 0.3$, following previous work (Li et al., 2025). When initially answering questions, we set the generation temperature to 0. Later, for sampling perturbed responses, we use a temperature of 1.0 to induce diversity. Finally, for each $q_i$, we sample $n = 20$ perturbed responses $\{r_i^{(j)}\}_{j=1}^n$ and embed them via $\{\mathbf{x_i^{(j)}} = E(r_i^{(j)})\}_{j=1}^n$, using the same embedding pipeline as in the external uncertainty setting.

## B.3 IMPLEMENTATION DETAILS

We set the PCA dimension to 15 and number of archetypes $K = 16$ for each benchmark. The Archetypal Analysis optimization is run for 2000 steps on each batch of sampled responses. For a fair comparison, both our method and the Semantic Volume baseline require an uncertainty threshold $\tau$ to make a binary prediction (hallucination vs. not) (Li et al., 2025). Following prior work, we tune this threshold on a held-out validation split (10% of the data), selecting the $\tau$ that maximizes the F1-score. Local uncertainty experiments are performed with $k = 5$ nearest neighbours for the local density metric.

**Baselines**  For Eccentricity and Degree (Lin et al., 2024), we adapt the code from previous work (`https://github.com/zlin7/UQ-NLG`) and use the entailment variant with the Deberta-Large model (`https://huggingface.co/microsoft/deberta-large` and an eigenvalue threshold of 0.7.

Table 3: Ablation over the number of sampled answers $n$ per question. For each subset we refit PCA on the subset only. We report mean±std AUROC/AUARC across runs and the mean $\Delta H$.

| $n$ | AUROC | AUARC | $\Delta H$ |
|---|---|---|---|
| 5 | $0.540 \pm 0.019$ | $0.576 \pm 0.017$ | 0.012 |
| 10 | $0.550 \pm 0.012$ | $0.593 \pm 0.015$ | 0.031 |
| 15 | $0.547 \pm 0.008$ | $0.587 \pm 0.019$ | 0.039 |
| 20 | $0.556 \pm 0.007$ | $0.585 \pm 0.019$ | 0.068 |

Table 4: Ablation over PCA dimension $d'$ at fixed $n = 20$. We fit a single PCA basis on all 20 answers and compare by truncation. We fix $K = 3$ so results remain feasible down to $d' = 2$.

| $d'$ | AUROC | AUARC | $\Delta H$ |
|---|---|---|---|
| 2 | $0.518 \pm 0.005$ | $0.564 \pm 0.017$ | $-0.021$ |
| 3 | $0.521 \pm 0.010$ | $0.567 \pm 0.015$ | $-0.022$ |
| 5 | $0.535 \pm 0.003$ | $0.575 \pm 0.017$ | 0.044 |
| 10 | $0.551 \pm 0.008$ | $0.581 \pm 0.019$ | 0.032 |
| 15 | $0.554 \pm 0.003$ | $0.585 \pm 0.018$ | 0.062 |

## C  ABLATIONS

### C.1  LOCAL UNCERTAINTY

We probe robustness of our local uncertainty estimator along three axes: number of sampled answers, PCA dimension, and number of archetypes.

First, Table 3 varies the number of sampled answers $n$. We maintain our fixed answer pool of 20 answers from main experiments: subsets of size $n$ are sampled uniformly without replacement from each question's fixed answer pool. When $n < n_{all}$ we evaluate $R = 30$ repeated subsets per question. We fit PCA on each subset, so the effective dimensionality obeys $d' \leq n - 1$. We scale the archetype count as $K = \text{round}(\rho n)$ with $\rho = K_{base}/n_{base}$ (16 / 20 for our chosen settings), then clip K to satisfy the assumptions of our theorem: $K \leq n$ and $K \leq d' + 1$, ensuring a non-degenerate, identifiable simplex in $d'$ dimensions. As expected, the hallucination correction rate increases with higher $n$, as the sampled set becomes more likely to contain one or more correct answers.

Second, Tables 4 and 5 vary the PCA dimension $d'$ at fixed $n = 20$: we show a fixed-$K$ view ($K = 3$) and a capacity-matched view ($K = \min(d' + 1, K_{base}, n)$) that respects the simplex capacity of a $d'$-dimensional space. In both cases, increasing $d'$ up to our chosen setting improves the hallucination detection performance. We suggest this is due to low PCA dimensions being unable to capture sufficient geometric details of the semantic answer space.

Finally, Table 6 sweeps the number of archetypes $K$ at fixed $d'$ (default $d' = 15$) while enforcing the same feasibility limits. Interestingly our local metric is able to perform well even with very low numbers of archetypes; we suggest this is due to two of the three component terms using the raw answer embeddings, such that altering the 'usage rarity' term does not drastically affect results.

**Statistics.**  For each setting we compute per–question metrics (AUROC, AUARC, and $\Delta H$) and then aggregate. In Table 3 (varying $n$), each question is evaluated on $R = 30$ i.i.d. subsets of size $n$ sampled uniformly without replacement from the fixed 20 answers; we refit PCA on each subset, compute the local score, and obtain the metric values per subset, then average across the $R$ subsets to get a single per–question estimate. Within a run, we then average across questions that pass the filter. Finally, the tables report mean±std across 3 independent runs. In Tables 4, 5 and 6 there is no within–question resampling (one evaluation per question); we average across questions within a run and report mean±std across runs.

Table 5: Ablation over PCA dimension $d'$ at fixed $n = 20$ with capacity-matched archetypes: $K = \min(d' + 1, K_{\text{base}}, n)$. This reflects the geometric limit that at most $d' + 1$ archetypes can be supported in a $d'$-dimensional space.

| $d'$ | AUROC | AUARC | $\Delta H$ |
|---|---|---|---|
| 2 | $0.519 \pm 0.003$ | $0.565 \pm 0.019$ | $-0.018$ |
| 3 | $0.520 \pm 0.007$ | $0.568 \pm 0.016$ | $0.016$ |
| 5 | $0.534 \pm 0.003$ | $0.573 \pm 0.020$ | $0.025$ |
| 10 | $0.552 \pm 0.005$ | $0.583 \pm 0.018$ | $0.052$ |
| 15 | $0.559 \pm 0.004$ | $0.586 \pm 0.021$ | $0.068$ |

Table 6: Ablation over the number of archetypes $K$ at fixed $n = 20$ and fixed $d'$ (default $d' = 15$). We enforce feasibility $K \le n$ and $K \le d' + 1$.

| $K$ | AUROC | AUARC | $\Delta H$ |
|---|---|---|---|
| 5 | $0.555 \pm 0.007$ | $0.584 \pm 0.020$ | $0.046$ |
| 8 | $0.556 \pm 0.003$ | $0.585 \pm 0.019$ | $0.063$ |
| 12 | $0.556 \pm 0.003$ | $0.586 \pm 0.020$ | $0.054$ |
| 16 | $0.556 \pm 0.003$ | $0.585 \pm 0.020$ | $0.064$ |

## C.2 GLOBAL UNCERTAINTY

We report similar ablations for global uncertainty in Table 7, showing similarly that high $n$ and $d'$ are desirable, with performance fairly constant as number of archetypes are varied. We choose $K = 16$ in our main results for improved performance in our local metrics.

Table 7: **Sensitivity Analysis on TriviaQA.** We report Mean $\pm$ Std over 3 runs. **(a)** As expected, increasing the sample size leads to improved performance. **(b)** The method requires $d' \ge 5$ to capture semantic structure; the poor performance at $d' = 2$ validates the necessity of high-dimensional volume over simple 2D area. **(c)** Geometric Volume is robust to the number of archetypes $K$, remaining stable even with a minimal simplex ($K = 4$).

| (a) Sample Size ($n$) | | | (b) PCA Dim ($d'$) | | | (c) Archetypes ($K$) | | |
|---|---|---|---|---|---|---|---|---|
| $n$ | AUROC | F1 | $d'$ | AUROC | F1 | $K$ | AUROC | F1 |
| 5 | $0.705 \pm 0.010$ | $0.652 \pm 0.017$ | 2 | $0.692 \pm 0.014$ | $0.666 \pm 0.002$ | 4 | $\mathbf{0.753} \pm 0.001$ | $0.706 \pm 0.001$ |
| 10 | $0.731 \pm 0.004$ | $0.674 \pm 0.012$ | 5 | $0.746 \pm 0.005$ | $0.701 \pm 0.007$ | 8 | $\mathbf{0.753} \pm 0.002$ | $0.704 \pm 0.004$ |
| 15 | $0.744 \pm 0.003$ | $0.682 \pm 0.014$ | 10 | $0.752 \pm 0.003$ | $\mathbf{0.705} \pm 0.004$ | 12 | $\mathbf{0.753} \pm 0.002$ | $\mathbf{0.707} \pm 0.002$ |
| 20 | $\mathbf{0.752} \pm 0.002$ | $\mathbf{0.701} \pm 0.004$ | 15 | $\mathbf{0.753} \pm 0.002$ | $\mathbf{0.705} \pm 0.002$ | 16 | $\mathbf{0.753} \pm 0.002$ | $0.705 \pm 0.004$ |

## D SELECTION OF GEOMETRIC SUSPICION TERMS

Our Geometric Suspicion score is composed of three separate terms. To determine a suitable choice for these terms, we conducted an analysis of our datasets generated for the global uncertainty evaluation.

For each model $M$ and dataset $D$ investigated, we have a set of $N_{D,M}$ questions. For each question, we have a 'default' answer, sampled with a temperature $T = 0$, and a corresponding hallucination label as determined by a judge LLM (as described in Appendix Section B). For each question, we have in addition $n = 20$ answers sampled with temperature $T = 1$, for which we generate hallucination labels as for the default answer. For each batch of answers, we then have the objects derived from our archetypal analysis: the answer embeddings $\mathbf{X}$, the convex decomposition coefficients $\mathbf{A}$, and the archetype embeddings $\mathbf{Z}$.

The primary intended application of our local uncertainty metric is hallucination reduction, so we first look at cases where there is a possibility to 'flip' a model response to a non-hallucination by preferential selection of one of the batch responses. Figure 4 plots a histogram of the sampled

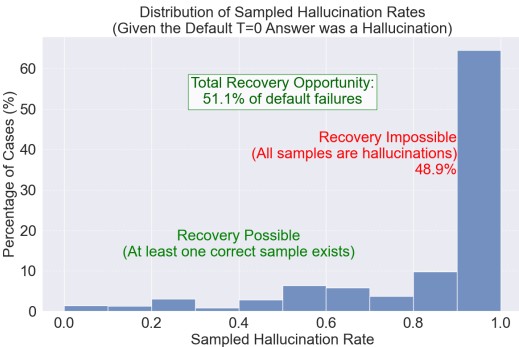

Figure 4: Distribution across all models and datasets of $T = 1$ hallucination rate when the $T = 0$ answer is hallucination.

hallucination rate to determine the opportunity for doing so within our datasets. In the vast majority of cases, when the model hallucinates in the default answer, a high proportion of the sampled answers are also hallucinations. Approximately half the time, however, at least one answer is a non-hallucination, providing an opportunity for an informed local metric to be of use.

Next, we aimed to find some high level characteristics of hallucinations within our response set. We selected cases where the default response was judged to be a hallucination, but at least one hallucination and non-hallucination was present in the batch of $T = 1$ answers. On the basic principle that for simple QA datasets there are many ways to be wrong, and fewer ways to be right, we formulated two hypotheses regarding the nature of hallucinations within the context of the embedding space spanned by each batch of answers. Firstly, hallucinations should sit in sparser regions of the embedding space compared to non-hallucinations. Secondly, hallucinations should sit closer to the edges of the batch embedding space, and therefore closer to the archetypes.

To test this, we compiled several terms of interest and compared them across our datasets. To measure the density of points within the embedding space, we used Local Density with $k = 5$ (as described in Section 3.4) and Voronoi cell volume. The Voronoi cell of an embedding $\mathbf{x}_i$ is the region of space closer to $\mathbf{x}_i$ than to any other point in the batch. A larger cell volume implies the response lies in a sparser region. As direct computation of these volumes in high-dimensional space is intractable, we approximate them via the dual Delaunay triangulation. First, for each batch of embeddings $\mathbf{X} = \{\mathbf{x}_1, \ldots, \mathbf{x}_N\}$, we reduce their dimensionality to a low-dimensional space (typically 3D) using Principal Component Analysis (PCA). We then compute the Delaunay triangulation of these reduced points. The volume of a response's Voronoi cell is approximated by summing the volumes of all Delaunay simplices that have the response's embedding as a vertex. A larger estimated volume signifies that a response lies in a sparser region of the local embedding space and therefore might be considered more suspicious.

To assess global position within the embedding space, we used the Distance from Consensus term described in Section 3.4 and Distance to Closest Archetype. The latter assesses a response's proximity to the boundaries of the semantic space spanned by the batch. For each response embedding $\mathbf{x}_i$, we compute its Euclidean distance to every archetype embedding $\mathbf{z}_k$, and take the minimum. A smaller distance means the response is closer to the edges of the batch semantic space and indicates greater suspicion. To align this with our other metrics where higher values denote higher suspicion, the final score is the negative of this minimum distance, defined as:

$$D_A(r_i) = - \min_{k \in \{1, \ldots, K\}} \|\mathbf{x}_i - \mathbf{z}_\mathbf{k}\|_2 \tag{16}$$

To assess the archetypal composition, we used Usage Rarity, as described in Section 3.4 and Geometric Entropy. The latter is the Shannon entropy (Shannon, 1948) of the archetypal reconstruction coefficients. The $i$-th row of $\mathbf{A}$, denoted $\boldsymbol{\alpha}_i = [A_{i1}, \ldots, A_{iK}]$, is a probability distribution describing how the pre-processed response embedding $\mathbf{x}_i$ is reconstructed from the set of learned archetypes $\mathbf{Z}$. A response that aligns clearly with a single archetype will have a sparse, low-entropy $\boldsymbol{\alpha}_i$, while a response that lies between multiple archetypes will have a more uniform, high-entropy $\boldsymbol{\alpha}_i$.

The Geometric Entropy for an individual response $r_i$ is defined as:

$$H_L(r_i) = H(\boldsymbol{\alpha}_i) = -\sum_{k=1}^{K} A_{ik} \log A_{ik} \tag{17}$$

Additionally, we split our dataset into subsets based on the sampled hallucination rate. As behaviour such as clustering could be very different depending on whether 1 or 19 of the 20 responses were hallucinations, we split the dataset into *low*, *mid-low*, *mid-high*, and *high* hallucination subsets, with the corresponding rates described in Table 8.

Table 8: Definition of sampled hallucination rate subsets used for granular analysis. The rate, $r$, refers to the fraction of hallucinated responses within a 20-sample set for a given question.

| Subset Name | Sampled Hallucination Rate ($r$) |
|---|---|
| Low | $0 < r \leq 0.25$ |
| Mid-Low | $0.25 < r \leq 0.50$ |
| Mid-High | $0.50 < r \leq 0.75$ |
| High | $0.75 < r < 1.0$ |

For each term and subset of interest, we then compared the distribution of term values for hallucinations versus non-hallucinations in the sampled answers. In calculating each term, we constructed the sign such that our hypothesis for its predictive value aligned with our local uncertainty framework: a high term value should lead to high suspicion of being a hallucination. Finally, we performed a one-sided Mann-Whitney U Test test to determine whether the term scores for hallucinations were significantly greater than for non-hallucinations. Table 9 shows the results.

Table 9: One-sided p-values from the Mann-Whitney U test, testing if metric scores for hallucinations are stochastically greater than for non-hallucinations across the defined data subsets (see Table 8). An extremely small p-value (e.g., $< 10^{-100}$) indicates a metric is both statistically significant and directionally correct. Values of $1.000$ indicate the metric is significant in the opposite direction (lower scores for hallucinations).

| Uncertainty Metric | Sampled Hallucination Rate Subset | | | |
|---|---|---|---|---|
| | **Low** | **Mid-Low** | **Mid-High** | **High** |
| Distance Consensus | $< 10^{-100}$ | $< 10^{-100}$ | 1.000 | 1.000 |
| Local Density | $< 10^{-100}$ | $< 10^{-100}$ | 0.0152 | 1.000 |
| Usage Rarity | $< 10^{-100}$ | $< 10^{-100}$ | 0.0544 | 1.000 |
| Voronoi Volume | $< 10^{-100}$ | $< 10^{-100}$ | 0.0739 | 1.000 |
| Geometric Entropy | $< 10^{-100}$ | $< 10^{-100}$ | 0.2903 | 1.000 |
| Distance Nearest Archetype | $< 10^{-100}$ | $< 10^{-100}$ | 0.2533 | 1.000 |

For all of our terms of interest, and hallucination rates less than 0.5, our intuitions were proved correct: non-hallucinations lie in denser regions of semantic space, are closer to a global consensus, and generally lie far away from archetypes and the edge of the embedding space. For high hallucination rates (over 0.75) this behaviour is completely reversed. We accept that in these cases it may be extremely hard to find the needle non-hallucination in a haystack of hallucinations. We suggest that in such situations it may be preferential to first use a global uncertainty measure to identify that the batch has high semantic dispersion, and defer from answering, as described in Section 6.

For the mid-high subset, for all our terms of interest bar 'Distance from Consensus' there is inconclusive evidence that hallucinations have greater values, although Local Density and Usage Rarity have p-values of 5% or below. This suggests potential to diagnose hallucinations within the batch even at these high hallucination rates, using a combination of multiple terms.

We choose Local Density and Usage Rarity as components of Geometric Suspicion due to their relatively high performance in the Mann-Whitney U tests, and their alignment with our original intuitions. We further include Distance from Consensus based on empirical observations of multiple

clusters in some response batches, corresponding to both hallucinations and non-hallucinations. In these cases we find the Distance from Consensus term helps to select the non-hallucinating cluster.

# E  LOCAL UNCERTAINTY EFFICACY

In Section 6, we note that the efficacy of local uncertainty measures inherently bounded by the quality of the sampled response set. To quantify this, we analyzed the relationship between the reduction in hallucination rate ($\Delta H$) and the proportion of correct answers available in the sampled batch for questions where the default response was a hallucination.

Figure 5 illustrates this relationship across all model and dataset combinations. The x-axis represents the *Potential to Correct*: defined as the median proportion of non-hallucinated answers found in the sampled batch given that the default greedy decode was incorrect. The y-axis shows the actual performance gain ($\Delta H$).

We observe a strong positive correlation. Notably, the case of Qwen3-8B on TriviaQA (the lowest performance in $\Delta H$ terms) demonstrates the 'confidently wrong' failure mode: while the model's default answer is incorrect, the sampled batch also contains very few correct alternatives. In such regimes, the semantic space is collapsed around the hallucination, and no re-ranking method can be effective. Conversely, as the correctness proportion of the sampled batch improves (moving right), Geometric Suspicion successfully identifies and retrieves the correct answers.

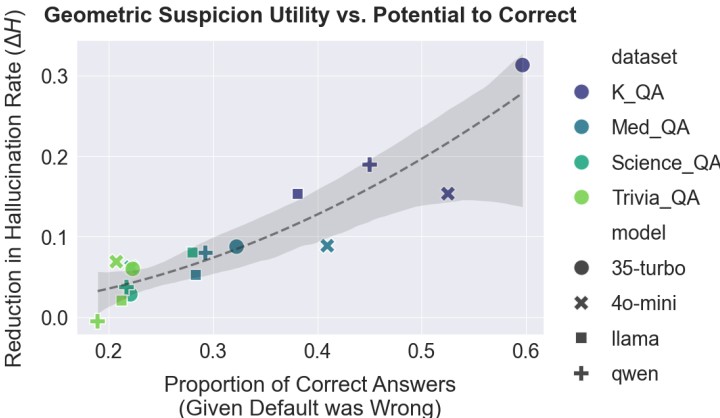

Figure 5: **Geometric Suspicion Utility vs. Corrective Potential.** The reduction in hallucination rate ($\Delta H$) is plotted against the median proportion of correct answers found in the sampled batch for instances where the default answer was incorrect. The dashed curve represents a second-order polynomial regression fit highlighting the general scaling trend, while the shaded region indicates the 95% confidence interval. Performance naturally drops in 'confidently wrong' regimes where the sampled batch lacks correct alternatives, but improves as the corrective potential increases.

# F  COMPUTATIONAL AND MEMORY COMPLEXITY

The dominant cost of our framework arises from the archetypal analysis step, which alternates between coefficient and archetype updates. Each update involves matrix multiplications over $n$ sampled responses of reduced dimension $d'$, resulting in a total time complexity of $\mathcal{O}(T\,nd'K)$, where $T$ is the number of alternating optimization steps and $K$ the number of archetypes. In our setting, $d' < n$ due to dimensionality reduction, and both $K$ and $T$ are small constants, making the method effectively linear in the number of responses and embedding dimensions.

The memory footprint is dominated by the response embeddings $\mathbf{X} \in \mathbb{R}^{n \times d'}$ and coefficient matrices $\mathbf{A} \in \mathbb{R}^{K \times n}$ and $\mathbf{B} \in \mathbb{R}^{n \times K}$, yielding an overall space complexity of $\mathcal{O}(nd' + nK + Kd')$. Since $K \leq \min(n, d' + 1)$ in all experiments, the approach remains both computationally and memory efficient for large-scale response batches.

