# OpenReview forum: "Geometric Uncertainty for Detecting and Correcting Hallucinations in LLMs"
_ICLR.cc/2026/Conference — Submitted to ICLR 2026_

### Official Review · Reviewer_nh9Z · 2025-10-27

**Soundness:** 3
**Presentation:** 2
**Contribution:** 2
**Rating:** 2
**Confidence:** 4

**Summary:**

This paper introduces a geometric framework for understanding uncertainty in large language models, distinguishing between global and local uncertainty. The authors propose a black-boxed method to both detect and mitigate hallucinations by modeling uncertainty from a holistic perspective. The paper evaluates the approach on several benchmarks, showing moderate performance improvements in some cases.

**Strengths:**

This paper proposes a novel and conceptually interesting perspective on hallucination detection and mitigation through geometric modeling of uncertainty. It introduces a distinction between global and local uncertainty, which offers a potentially useful framework for understanding LLM confidence. The paper also propose an appealing high-level idea to address hallucination detection and mitigation in a unified framework.

**Weaknesses:**

(1) My main concern is the lack of essential ablations. As a sampling based method, key hyperparameters such as sampling temperature and number of samples are not studied, but rather fixed to the same value through out the paper. This makes it difficult to assess the generalizability of the proposed approach.

(2) While the authors highlight the importance of medical data, the proposed method on MedicalQA underperforms baselines in AUROC on most models in Table 1.  Also, there's typo in the color code in table 1-- on CLAMBER dataset llama3.1 8B, performance of proposed method should not be highlighted for AUROC.

(3) Some simple baselines are missing. On the detection task on open-source model, how does the performance compared to simple perplexity based detection? On the mitigation task, how does the performance compared to majority vote among the generated answers? -- this is particularly interesting since the design of detection score is to find consensus.

**Questions:**

see weakness

---

> ### Author Response · Authors · 2025-11-20
> **Author response to reviewer nh9Z**
>
> 1. *'key hyperparameters such as sampling temperature and number of samples are not studied, but rather fixed to the same value through out the paper.*'
>
>     Thank you for pointing out this omission, we have added ablation studies across PCA dimension, number of archetypes, and number of samples. Following previous work in this field, we set temperature to 1 throughout and do not ablate. Temperatures in the range 0.7-1.0 are common defaults in web interfaces and APIs, and previous works in semantic dispersion-based uncertainty find values in this range to be effective.
>
> 2. '*While the authors highlight the importance of medical data, the proposed method on MedicalQA underperforms baselines in AUROC on most models in Table 1. Also, there's typo in the color code in table 1-- on CLAMBER dataset llama3.1 8B, performance of proposed method should not be highlighted for AUROC.*'
>
>     Thank you for spotting the typo, we have corrected this. We use two datasets, MedicalQA and K-QA, to assess performance on medical tasks, and report two metrics for each of four models. Our method performs best or joint best in 10 out of these 16 reported metrics, so we think it fair to highlight its performance in this domain.
>
> 4. *'On the detection task on open-source model, how does the performance compared to simple perplexity based detection? On the mitigation task, how does the performance compared to majority vote among the generated answers?'*
>
>     Thank you for pointing this out, we have added two baselines for the hallucination mitigation task, namely eccentricity and degree.
>
>     We have not added perplexity based detection since we aimed to compare to other black box methods that operate on the generated text alone. Perplexity requires access to token-level logits, which are not available in many API settings.
>
>     Majority vote is non-trivial to add for the open-ended tasks we consider since there is no constraint on answer format. We argue that the Degree baseline (Lin et al., 2024), which we have added to our local uncertainty experiments, serves as the robust, semantic proxy for Majority Vote in this context. Geometrically, the "majority" consensus corresponds to the region of highest semantic density. The Degree metric captures this by summing the similarity between a response and its neighbors. By comparing against Degree (and demonstrating superior performance with Geometric Suspicion), we show that our method outperforms a semantic majority vote strategy.

---

### Official Review · Reviewer_gaV1 · 2025-10-29

**Soundness:** 2
**Presentation:** 3
**Contribution:** 2
**Rating:** 2
**Confidence:** 4

**Summary:**

The authors propose a hallucination detection method that lies within a geometric framework that uses black-box model access. The framework includes both global uncertainty estimates through measuring geometric volume, while local uncertainty estimates use geometric suspicion. In doing this, the framework is able to capture both global and local uncertainty in the LLM responses, quantifying the reliability of the response through semantic boundary points.

**Strengths:**

- The theoretical framework (Appendix A)  to justify allows for a higher-level understanding of the ideas proposed in the paper. I would suggest you find a way to include this in the main paper.
- The benchmarks used to evaluate the framework are adequate and diverse. Although most are focused on medical data, there are more general purpose datasets used as well.

**Weaknesses:**

- The performance, as reported in table 1, is not consistently higher than other baselines. More experiments need to be conducted to understand why this is the case. It seems that P(True), the simplest baseline out of all of them, outperforms in certain scenarios, so considering the complexity associated with the proposed approach in comparison with P(True), there needs to be better justification by the others.
- A complexity and/or time analysis of the framework is missing from the paper.

**Questions:**

see above.

---

> ### Author Response · Authors · 2025-11-20
> **Author response to reviewer gaV1**
>
> ### Responses to strengths / weaknesses
>
> We can see that grounding our methods in the theoretical analysis makes for a clearer argument; we have moved the key points to the main paper in the Methods section, under subsection 'Simplicial Volume as a Proxy for Uncertainty'.
>
> ### Responses to questions
>
> 1. *'The performance, as reported in table 1, is not consistently higher than other baselines...It seems that P(True), the simplest baseline out of all of them, outperforms in certain scenarios, so considering the complexity associated with the proposed approach in comparison with P(True), there needs to be better justification by the others.'*
>
>     While we agree that P(True) is a strong baseline (particularly when bolstered by few-shot examples as in our implementation), we respectfully argue that comparing raw detection performance alone overlooks the distinct functional advantages and robustness offered by our geometric framework. The primary justification for our method's complexity is that it unifies global detection and local correction within a single framework; we have reworded our abstract and contributions to emphasize this.
>
>     P(True)is essentially a rejection mechanism: it provides a single scalar telling us whether to trust the model's output. If the score is low, the user is left with no alternative but to discard the response. Geometric Suspicion is a selection mechanism: By explicitly modeling the response manifold, our method allows us to not only flag uncertainty but actively recover the correct answer from a noisy batch (as demonstrated in our Best-of-N experiments).
>
>     We also note the finding from Farquhar et al. (2024) that 'P(True) seems to improve with model size, suggesting that it might become more competitive for very capable honest models in settings that the model understands well (which are, however, not the most important cases to have good uncertainty)'. Similarly, we see P(True) performing well for the likely larger GPT-4o-mini and GPT-3.5-Turbo models (two metric wins vs one).
>
>     Ultimately our method does not massively surpass existing baselines because it relies on the same principle of semantic dispersion in sampled answers predicting hallucination. We offer, we believe, a novel geometric framework to assess this dispersion (for global uncertainty), and sensible metrics to assess model confidence at the response level. These offerings are still constrained to settings where the model is 'uncertain and wrong', i.e. there is sufficient diversity in the sampled answers. There exist cases where the model is confident and wrong (no diversity in sampled answers), which we accept our method does not address and leave to future work. We add a paragraph in the 'Discussion' section regarding this point.
>
> 3. *'A complexity and/or time analysis of the framework is missing from the paper.'*
>
>     Thank you for pointing this out, we have added this in Appendix Section F.

---

> > ### Comment · Reviewer_gaV1 · 2025-11-22
> >
> > Thank you for the clarifications and for addressing the points raised in the review. The added theoretical context, complexity analysis, and expanded comparison with P(True) provide useful clarification. My assessment remains based on the originally submitted manuscript, and I will not be changing my score.

---

> > > ### Author Response · Authors · 2025-11-22
> > >
> > > May we politely ask you to reconsider changing your score if you feel that we have addressed your key concerns in our response and updated manuscript? The ICLR guidance to reviewers instructs to "Maintain a spirit of openness to changing your initial recommendation", so we don't think it is fair to disregard all changes made during this review process.
> > >
> > >
> > > If you have further issues with the paper please point them out so we can endeavour to address them.

---

### Official Review · Reviewer_nv83 · 2025-10-31

**Soundness:** 2
**Presentation:** 2
**Contribution:** 2
**Rating:** 4
**Confidence:** 5

**Summary:**

This manuscript presents a new framework to quantify the uncertainty of LLM responses from a geometric perspective. The framework provides both a group-wise uncertainty score and a response-wise uncertainty score based on archetypal analysis. Experiments on several QA benchmark are performed to validate the effectiveness of the approach.

**Strengths:**

1.	Using archetypal analysis to quantify the uncertainty of LLM responses provides a new perspective.
2.	Providing response-wise uncertainty is of more practical values than existing prompt-wise uncertainty metrics.

**Weaknesses:**

1.	The main claim “this is the first sampling-based black-box method that differentiates high and low uncertainty responses within a batch.” is not true. See [1], where the proposed semantic density metric also measures sampling-based response-wise confidence/uncertainty without accessing the internal LLM state.
2.	The design of the local uncertainty, i.e., the sum of ranks in three simple heuristic metrics, is not supported by any solid mathematical justifications.
3.	The interpretation of the experimental results need to be further clarified. See “Questions” below.
4. Several related baseline methods are missing in the current experiments. See “Questions” below for more details.

**Questions:**

1.	Can you add more discussions and compare to Semantic density [1], which is also a sampling-based response-wise confidence/uncertainty metric without access to the internal LLM state? Moreover, please also consider comparing to Degree [2] and the length-normalized likelihood [3].
2.	When converting the responses into embedding space, the prompt is not considered as a context. This may cause problems for measuring the semantic relationships among the responses. One example: Q: “What is the capital of France?” A1: “Paris”, A2: “the capital of France is Paris”. A1 and A2 actually mean the same thing under this context, but they will have very different embeddings without considering the original questions as context. Have you considered this limitation?
3.	In the "Convex Hull Approaches" part of “Related Works”, existing works are criticized to oversimplify the problem by using PCA to reduce the embedding dimensionality to two. However, the proposed approach also uses PCA to reduce the embedding dimension (to 15 dimensions). How do you make sure 15 dimensions are sufficient to preserve important information? Do you have an ablation study with different PCA dimensions?
4.	The likelihood of each sampled response is not utilized in the local uncertainty calculations. Using this information can potentially reduce the cost of sampling, i.e., we don’t need to sample the same response multiple times to estimate the output distribution. What is your consideration here?
5.	In the experimental setup, it is stated “to classify response sets as reliable or hallucinated…”. What do you exactly mean here? Are you using the default answer to represent the response set?
6.	Did you handle the long-form question/answer in medicalQA in a different way than other shorter QA benchmarks? If not, do you think one uncertainty score for the entire long answer, which may include multiple claims, is sufficient?
7.	Do you have an explanation or analysis about why the answer selection does not work well in Qwen3-8B? What makes the performance difference so different across different model?

[1] Xin Qiu, Risto Miikkulainen. Semantic density: Uncertainty quantification for large language models through confidence measurement in semantic space, Advances in Neural Information Processing Systems (NeurIPS), 2024

[2] Zhen Lin, Shubhendu Trivedi, Jimeng Sun, Generating with Confidence: Uncertainty Quantification for Black-box Large Language Models, Transactions on Machine Learning Research, 2024

[3] Kenton Murray, David Chiang. Correcting Length Bias in Neural Machine Translation. In Proceedings of the Third Conference on Machine Translation, 2018.

---

> ### Author Response · Authors · 2025-11-20
> **Author response to weaknesses**
>
> > The main claim “this is the first sampling-based black-box method that differentiates high and low uncertainty responses within a batch.” is not true. See [1], where the proposed semantic density metric also measures sampling-based response-wise confidence/uncertainty without accessing the internal LLM state.
>
> We would argue  that methods requiring sequence likelihoods, like semantic density, are not black box. In the case of [2] however, we agree these methods are black box, and we add them to our baseline experiments (Eccentricity, Degree). We additionally note in the Methods and Discussion that these two metrics are highly related to components of our 'Geometric Suspicion' metric.
>
> We can see that our claim regarding being the first black-box method for local uncertainty requires adjustment. We update our abstract and introduction to emphasize the novelty of our method.
>
> In the abstract:
>
> > While recent black-box approaches have shown some success, they often rely on disjoint heuristics or graph-theoretic approximations that lack a unified geometric interpretation...Unlike prior methods that rely on discrete pairwise comparisons, our approach provides continuous semantic boundary points which have utility for attributing reliability to individual responses.
>
> And in the introduction:
>
> > ...our method offers a geometrically grounded alternative to heuristic graph measures for differentiating high and low uncertainty responses.
>
>
> We emphasize that our primary contribution is a unified geometric framework. While Lin et al. utilize discrete graph-theoretic heuristics (pairwise edges) to approximate local confidence, our method models the semantic space as a continuous convex hull defined by Archetypes. This allows us to derive Global Uncertainty (Volume) and Local Uncertainty (Suspicion) from the same underlying mathematical object. Furthermore, our inclusion of 'Usage Rarity' leverages the unique properties of archetypal reconstruction, which has no direct analogue in standard graph-based approaches.
>
> > The design of the local uncertainty, i.e., the sum of ranks in three simple heuristic metrics, is not supported by any solid mathematical justifications
>
> We argue that our local uncertainty measure has justification in the context of approximating a probability distribution over the semantic space spanned by the convex hull volume. Like previous confidence measures (e.g. Semantic Density, Degree, Eccentricity), our metric is based on intuitions regarding this answer distribution, rather than derived purely from mathematical principles. We have however moved Theorem 1 from the Appendix to the methods section, to emphasize this link between our method and semantic distributions.
>
> The components of our metric are related to similar metrics proven effective in previous works; we update the Methods section to acknowledge this, adding a paragraph beginning 'Relation to prior local UQ baselines and intuition'. We accept however that the rank sum combination of components has poor statistical footing and replace it with a Fisher combination of empirical p-values.

---

> > ### Author Response · Authors · 2025-11-20
> > **Author response to questions**
> >
> > 1. '*Can you add more discussions and compare to Semantic density, ... Degree and the length-normalized likelihood'*
> >
> >     Thank you for drawing our attention to these works; we can see they are highly relevant and have added reference to them in 'Related Works' in a 'Semantic Space Confidence' paragraph. We would argue however that methods requiring sequence likelihoods (including Semantic Density and length-normalized likelihoods) are not black box, so we intentionally do not compare to this type of baseline in our experiments.
> >
> >
> >
> > 2. *'When converting the responses into embedding space, the prompt is not considered as a context... Have you considered this limitation?'*
> >
> >     We carefully considered whether to concatenate the prompt with the response but decided against it for several reasons. Firstly, we align our approach with standard practices in black-box uncertainty quantification (e.g., Semantic Entropy [Farquhar et al., 2024]), which consistently embed responses in isolation to maximize the sensitivity to variations in the output distribution. Secondly, especially for short-form QA where the question is longer than the answer, concatenating the prompt acts effectively as adding a large constant to every response vector. Intuitively this reduces the differences between responses and dampens the dispersion signal we rely on to detect hallucinations. Finally, we utilize gte-Qwen2-1.5B-instruct, a modern instruction-tuned embedding model. Unlike older BERT-based models, these encoders are specifically trained on Semantic Textual Similarity (STS) tasks to map structurally different but semantically equivalent sentences (e.g., "Paris" vs. "The capital is Paris") to nearby points in the embedding space, even without the context of the prompt.
> >
> > 3. *'How do you make sure 15 dimensions are sufficient to preserve important information? Do you have an ablation study with different PCA dimensions?'*
> >
> >     Thank you for pointing out this omission, we have added an ablation study in Appendix Section C showing that 15 dimensions are sufficient and appropriate for our problem setup. In particular, we show that decreasing PCA dimension to 2 leads to poor hallucination detection via our global uncertainty metric, and ineffective hallucination correction for local uncertainty.
> >
> >
> > 4. *'The likelihood of each sampled response is not utilized in the local uncertainty calculations.'*
> >
> >     We agree that using likelihoods could help reduce sampling costs, among other things, however we aimed to create a truly black box method which uses only the text responses from the model.
> >
> > 5. *'In the experimental setup, it is stated “to classify response sets as reliable or hallucinated…”. What do you exactly mean here? Are you using the default answer to represent the response set?'*
> >
> >     We mean the converse, that we are using the response set to 'represent' or make judgements about the default answer. We have clarified this statement in the Results section as follows:
> >
> >     > ...we test Geometric Volume’s ability to classify $r_\text{default}$ as reliable or hallucinated given a response set
> >
> > 6. *'Did you handle the long-form question/answer in medicalQA in a different way than other shorter QA benchmarks? If not, do you think one uncertainty score for the entire long answer, which may include multiple claims, is sufficient?'*
> >
> >     We used the same methods for all benchmarks. We accept that our method is reliant on the capability of a single embedding vector to represent the semantic content of an answer. As answers become longer, this assumption may break down, although recent work (Semantic Isotropy, referenced in Discussion section) has shown the single embedding method holds promise for answers up to 1000 words. We have added a short paragraph addressing this limitation in the Discussion section.
> >
> > 7. *'Do you have an explanation or analysis about why the answer selection does not work well in Qwen3-8B?'*
> >
> >     We have added some short analysis and discussion about this point in the Results, Discussion and Appendix Section E. The performance of preferential answer selection is linked to the rarity of a correct answer within the sampled set: a set with only one correct answer out of 20 is a harder problem than one with five correct answers. In the particular case of Qwen3-8B and TriviaQA, the model is in general 'confidently wrong': when the default answer is a hallucination, the sampled set contains few correct answers. Appendix Section E contains a plot with the associated evidence.

---

### Author Response · Authors · 2025-11-20
**Author summary response to all reviewers**

We thank all reviewers for their feedback, and note the consensus that our work represents a "new", "conceptually interesting" perspective for LLM uncertainty modelling, whilst offering "practical value" with our hallucination mitigation application. We are also pleased to see reviewer support for the unified framework for prompt-wise and response-wise uncertainty measures, and the diversity of our benchmark datasets.

The review process has been highly productive. In particular, the discussion regarding prior graph-based baselines (e.g., Degree and Eccentricity) has helped us sharpen our core contribution: moving from a claim of "the first black-box local method" to "the first unified geometric framework" that derives both global dispersion and local reliability from a single continuous space.

We have updated the manuscript to reflect this refined positioning and included comprehensive new experiments to address reviewer requests:

1. **Addition of baselines**: We implemented and compared against Degree and Eccentricity. These experiments confirm that while graph heuristics perform well in some settings, our Geometric Suspicion method demonstrates superior robustness, particularly in complex domains like MedicalQA.
2. **Positioning and grounding**: We revised the Abstract and Introduction to explicitly contrast our unified continuous framework against prior graph-theoretic approaches. We moved our key theorem (linking convex hull volume to entropy) from the Appendix to the Methods section to better highlight the theoretical justification of our approach.
3. **Ablations**: We added ablations for the key hyperparameters in our framework, justifying our setting selection.
4. **Further Analysis**: We added experiments investigating the efficacy of our hallucination correction mechanism across models and datasets. We also add complexity analysis confirming our method is computationally feasible.

We reply to all reviewers individually below.

---

### Author Response · Authors · 2025-12-02
**Author Summary for new Area Chair**

In our earlier comment ***'Author summary response to all reviewers'***, we detailed the updates made to the manuscript during the discussion phase. In light of the changes to the ICLR review process, we would like to briefly restate the case for acceptance and how the paper has evolved in response to reviewer feedback.


We underline again the reviewer consensus that our work represents a "new", "conceptually interesting" perspective for LLM uncertainty modelling, whilst offering "practical value" with our hallucination mitigation application. We have since worked to strengthen the paper based on reviewer comments and questions.

 - Positioning and theory: We refined our main claim to “a unified geometric framework” for global and local uncertainty, clarified the relationship to prior graph-based methods (including Degree and Eccentricity), and moved key theoretical results into the main Methods section.

 - Baselines and related work: We added Degree and Eccentricity as local UQ baselines and clarified the relationship to semantic density and likelihood-based methods, while explaining our focus on strictly black-box approaches.

- Ablations and analysis: We added ablations for PCA dimension, number of archetypes, and number of samples, provided complexity analysis, and expanded discussion of model- and dataset-specific behaviour.

Reviewer nv83 raised the most detailed set of questions; we believe we have responded comprehensively to each point with new experiments, clearer positioning, and additional analysis. The concerns of the other reviewers focused primarily on baselines, ablations, and complexity, which we have also directly addressed.

Because reviewer scores are now fixed to their pre-discussion values, they cannot reflect these substantive revisions. We respectfully ask that you evaluate the paper in light of (i) the positive aspects identified in the original reviews and (ii) the concrete changes documented in our rebuttals and updated manuscript.

---

### Meta-Review · Area_Chair_p4pu · 2026-01-08

**Summary:**

The paper proposes a geometric uncertainty quantification method using archetypal analysis over sampled answer embeddings: a global geometric volume (convex-hull volume of archetypes) for uncertainty quantification and a local geometric suspicion (density/consensus/rarity features) for best-of-N selection. While the paper contains interesting ideas, the current submission does not convincingly establish novelty and empirical advantage over existing uncertainty quantification methods for LLMs.

In particular, reviewers hold concerns over the novelty of the paper, have reservations about the heuristic approach, and have pauses about the mixed empirical results. As a result, the submission in its current form falls short of an ICLR paper.

**Reviewer Concerns:**

Reviewers noted that response-wise uncertainty from sampled generations has been studied previously (e.g., semantic-density style uncertainty quantificatoin), and the paper’s original “first black-box” positioning was viewed as overstated; the rebuttal appropriately repositions the contribution as a unifying framework, but the closest comparisons and positioning remain incomplete. Methodologically, the local score is still largely heuristic (an aggregation of density/consensus/rarity signals), and although the rebuttal improves the statistical combination and adds additional baselines/ablations, the core design choices are not sufficiently justified. Empirically, results appear mixed across datasets and models, with some settings where simple baselines (e.g., self-evaluation/P(True) or graph/density heuristics) match or outperform the proposed approach, making it hard to justify the additional complexity and sampling cost for a generally applicable method; concerns about missing key ablations (notably temperature sensitivity) and evaluation choices (including reliance on LLM-as-judge in some regimes) further limit confidence.

**Reviewer Scores:**

It's unlikely that the reviews will be flipped enough to make this submission an accept.

---

### Decision · Program_Chairs · 2026-01-26

Reject